# Anti-Inflammatory and Anti-Bacterial Effects of Mouthwashes in Intensive Care Units: A Systematic Review and Meta-Analysis

**DOI:** 10.3390/ijerph20010733

**Published:** 2022-12-30

**Authors:** Yong Xiang, Xiaolin Ren, Ye Xu, Li Cheng, He Cai, Tao Hu

**Affiliations:** State Key Laboratory of Oral Diseases, National Clinical Research Center for Oral Diseases, Department of Preventive Dentistry, West China College of Stomatology, Sichuan University, No. 14, 3rd Section, Renmin South Road, Chengdu 610041, China

**Keywords:** critical care, anti-inflammation, antibacterial agents, intensive care unit, mouthwashes, oral care

## Abstract

Mouthwashes are used as oral care for critical care patients to prevent infections. However, there are conflicting data concerning whether mouthwashes are needed as a part of daily oral care for critical care patients. This study aimed to evaluate the anti-inflammatory and anti-bacterial effects of mouthwashes for critical care patients. The PubMed, EMBASE, CENTRAL, and grey literature databases were searched by descriptors combining population (intensive care unit patients) and intervention (mouthwashes). After the screening, only randomized controlled trials (RCTs) evaluating the anti-inflammatory and anti-bacterial effects of mouthwashes in patient critical care were included. From the 1531 articles, 16 RCTs satisfied the eligibility criteria for systematic review and 10 were included in the meta-analyses. A significant difference was found in the incidence of ventilator associated pneumonia (VAP) (odds ratio [OR] 0.53, 95% confidential interval [95% CI] 0.33 to 0.86) between the mouthwash and placebo groups, while no significant difference was found in the mortality (OR 1.49, 95%CI 0.92 to 2.40); the duration of mechanical ventilation (weighted mean difference [WMD] −0.10, 95%CI −2.01 to 1.81); and the colonization of *Staphylococcus aureus* (OR 0.88, 95%CI 0.34 to 2.30), *Escherichia coli* (OR 1.19, 95%CI 0.50 to 2.82), and *Pseudomonas aeruginosa* (OR 1.16, 95%CI 0.27 to 4.91) between the two groups. In conclusion, mouthwashes were effective in decreasing the incidence of VAP. Thus, mouthwashes can be used as part of daily oral care for critical care patients.

## 1. Introduction

Patients admitted to intensive care units (ICUs) generally find it difficult to maintain daily oral hygiene due to the use of mechanical ventilation [1]. In addition, considering that any discomfort or dislodgement of the endotracheal and orogastric tubes may increase the risk of accident, nurses are usually unlikely to carry out thorough tooth brushing for critical care patients [2,3,4]. As a result, bacteria accumulate rapidly and the colonization of microbial pathogens may follow [5,6,7], which can impair the normal defense mechanisms of critical care patients for resisting infection and this ultimately leads to some unpleasant complications [8,9].

One of the most common and serious complications that may develop in patients in ICUs is ventilator associated pneumonia (VAP) [1]. Studies have reported that the risk of VAP progress varies between 10% and 25% within the first 48 h after intubation [10]. Furthermore, VAP is considered to be a crucial cause of a prolonged ICU stay and an increase in morbidity, mortality and health costs [11,12,13]. In addition, the development of other complications, such as periodontitis, may increase the occurrence of systemic diseases, e.g., atherosclerosis, pneumonia, and cancer [14,15]. As the primary burden of these complications, it is essential to develop preventive approaches in ICUs.

To reduce the incidence of infectious complications, various forms of antibiotic prophylaxis have been applied for patients in ICUs. A study [16] has found that selective digestive decontamination as a prophylactic antibiotic strategy was beneficial to improve clinical outcomes for critical care patients. However, the increased risk of the induction and selection of resistant pathogens has limited the use of antibiotics as routine prophylaxis [10]. Moreover, the loss of natural physiological characteristics and normal oral flora may result in harmful reactions, rather than beneficial reactions [17]. Innovative approaches to critical care are needed for the prevention of infection.

Mouthwashes have been recommended as regular oral care for critical care patients to prevent infections [18,19], as they could augment the local concentration [20], decrease the cost, and reduce plaque formation [21]. Furthermore, studies have indicated that mouthwashes were effective in reducing the incidence of VAP [4,22] and the colonization of *Staphylococcus aureus* [23,24]. In addition, multiple types of mouthwashes, such as chlorhexidine [17,25], hydrogen peroxide [22] and sodium bicarbonate [26], have been used in critical care patients. However, some studies have reported that mouthwashes may increase the risk of mortality and cause some other side-effects [27,28,29]. There are still conflicting data concerning whether mouthwashes are needed as a part of daily oral care for critical care patients. Although some meta-analyses have explored the efficacy of chlorhexidine mouthwashes on preventing VAP [1,30], no systematic review has comprehensively evaluated the anti-inflammatory and anti-bacterial effects of mouthwashes for critical care patients, particularly on the colonization of pathogenic bacteria. Thus, the aim of this systematic review is to evaluate both the anti-inflammatory and anti-bacterial effects of mouthwashes for patients admitted to ICUs in order to provide relevant evidence for clinical decision making and practice on patient critical care.

## 2. Materials and Methods

A detailed protocol was developed a priori and registered in the International Prospective Register of Systematic Reviews (http://www.crd.york.ac.uk/PROSPERO/ accessed on 18 March 2019) (registration number: CRD42019127016). The present systematic review followed the Preferred Reporting Items for Systematic Reviews and Meta-analyses (PRISMA) statement and checklist.

### 2.1. The Population-Interventions-Comparisons-Outcomes-Study Design (PICOS) Question

The current systematic review was conducted to answer the focused question: “*How are the effects of mouthwashes for inflammatory and bacterial control as a supplement of daily oral care when compared to control group among the patients admitted to ICUs?*” in accordance with the following PICOS elements:Population: patients admitted to ICUs with endotracheal tubes or mechanical ventilation.Intervention(s): application of mouthwashes.Comparison(s): application of placebos, no intervention, or usual care.Outcomes: both the anti-inflammatory and anti-bacterial efficacy of mouthwashes.Study design: only randomized controlled trials (RCTs).

### 2.2. Eligibility Criteria

According to the above PICOS question, studies were eligible for this systematic review if they fulfilled the following inclusion criteria:The participants of studies should be admitted to ICUs with endotracheal tubes or mechanical ventilation.The intervention group(s) should use mouthwashes by liquids, sprays, or with a swab at least once a day as part of their oral care.The comparison group(s) should include placebos, no intervention or usual care.Studies should include at least one inflammatory or bacterial parameter as the outcome measure, e.g., the VAP incidence, periodontal conditions, bacterial colonization, etc.The studies should be RCTs.Abstract and full text available.Studies written in English.

Studies using duplicate data from the same cohorts; studies without specific description; studies employing combinations of various interventions in which the effects of mouthwashes could not be distinguished from other components; studies did not concentrate on oral care but other ways (e.g., bathing, hand-washing); studies without randomization; and uncontrolled studies, observational studies, in vitro studies, animal studies, cadaver studies, case studies, letters and historical reviews were excluded.

### 2.3. Information Sources and Literature Search

The PubMed, EMBASE, and Cochrane Central Register of Controlled Trials (CENTRAL) databases were systematically searched for relevant articles written in English, from the inception of the database to 29 March 2022. To gain a highly sensitive group of descriptors, the search strategy was set up on the PICOS framework, and combined the population- and intervention-related MeSH and free text words.

(1)Population- (“Intensive care units” [MeSH] OR “intensive care” OR “critical care”)(2)Intervention- (Mouthwashes OR mouthrinse* OR mouthwash* OR ((rinse* OR wash* OR gargle* OR collut*) AND (oral* OR dent* OR mouth*)))(3)Population AND Intervention–#1 AND #2

For unpublished works, “grey” literature databases were also searched following the above search strategy, combining population (intensive care unit patients) and intervention (mouthwashes). For unpublished clinical trials or registries, ClinicalTrials.gov (https://beta.clinicaltrials.gov/ accessed on 29 March 2022) and the International Clinical Trials Registry Platform (https://www.who.int/clinical-trials-registry-platform accessed on 29 March 2022) were searched. For relevant dissertations and theses, the ProQuest Dissertation Abstracts and Thesis database (https://about.proquest.com/ accessed on 29 March 2022) was searched. The Web of Science (http://www.webofscience.com accessed on 29 March 2022) was searched by the Conference Proceedings Citation Index-Science for conference proceedings. Additionally, some other online resources, such as the GreyNet databases (https://easy.dans.knaw.nl/ui/home accessed on 29 March 2022), were also searched as a supplement. Furthermore, a manual search was conducted according to the reference lists of included studies for any further relevant studies.

### 2.4. Study Selection

After searching, all returned citations were downloaded and imported to the Endnote version X9 software (The Clarivate Analytics Ltd., Beijing, China) for study selection. Duplicates were removed, and the title and abstract of each item were screened by two independent reviewers (Y.Xiang. and X.Ren.). After selecting excluded items, full texts of the potentially relevant articles were retrieved. The same two reviewers screened the above full texts articles, respectively. During the study selection, articles that failed to meet the criteria were excluded immediately. Discordances regarding the procedure were resolved through discussion or by a third reviewer (H.Cai.).

### 2.5. Data Collection and Data Items

A data extraction table was developed *a priori*, and the data were extracted by two independent reviewers (Y.Xiang. and X.Ren.). Specific data pertaining to: first author and year of publication; sample size; average age of population; number of males and smokers; types, dosage and frequency of applied mouthwashes; comparison(s); devices for applying mouthwashes; type(s) of outcome measure(s); follow-up; dropout; adverse events; and the final effects of mouthwashes were extracted. Next, the extracted data were cross-checked for accuracy and agreed by the same two assessors (Y.Xiang. and X.Ren.). Any disagreement was resolved by discussion or, if necessary, consulting a third assessor (H.Cai.).

### 2.6. Risk of Bias in Individual Trials

Two reviewers (Y.Xiang, X.Ren.) graded the risk of bias for all the included studies. Version 2 of the Cochrane risk-of-bias tool (RoB2) was used for assessing the risk of bias of the included RCTs [31]. The risk of bias of each RCT was estimated as “low”, “some concerns” or “high” through five domains, including the randomization process, deviations from intended interventions, missing outcome data, measurement of outcome, and selection of the reported result.

### 2.7. Summary Measures

The anti-inflammatory effects of the interventions and comparisons in the included studies were evaluated by outcome measures, such as the incidence of VAP, the mortality, and the duration of mechanical ventilation. In addition, the anti-bacterial effects were commonly assessed by the colonization of bacteria. A few studies reported the oral inflammation by different measures, such as the Gingival Index and Beck Oral Assessment Scale. For further meta-analyses, the relevant outcome data was collected for those measures in varied formats (e.g., means, standard deviations (SDs), standard errors, medians, and interquartile ranges).

### 2.8. Synthesis of Results and Sensitivity Analysis

To assess the anti-inflammatory and anti-microbial effects of mouthwashes for critical care patients and to increase the precision of the effect size of the overall estimate, the data of sufficient studies (n≥ three) with the same comparison and the same outcome measure were extracted to conduct meta-analyses. Owing to the random distribution in individual studies, the events of the outcome at the endpoint of the research and the sample size in the intervention and comparison groups were obtained to estimate the odds ratios (OR) and 95% confidential intervals (95% CIs) between the groups, and the mean and SD were obtained to estimate the weighted mean differences (WMDs) and 95% CIs, thus assessing the anti-inflammatory and anti-microbial effects of mouthwashes. A few articles reported other data formats and, if possible, were transformed into the ideal format (e.g., means and SDs) to be included in the analyses. The average SDs from the studies in the same meta-analyses was used to estimate the missing SDs in some studies. The data from the included studies were analyzed using RevMan version 5.3 software (Cochrane Collaboration, Copenhagen, Denmark). The heterogeneity across the studies was estimated using the Chi-squared test and I^2^ statistics. In consideration of the sample size of each study and the heterogeneity across the studies, a random-effects model or a fixed-effects model was applied. Sensitivity analyses were performed by excluding each study from the analyses to assess the potential influence of unknown confounding. No meta-regression or subgroup analysis was performed as a result of the limited number of eligible studies.

### 2.9. Risk of Bias across Studies

The Egger’s test was used to evaluate the publication bias across the studies using Stata/SE version 16 (StataCorp, College Station, TX, USA).

## 3. Results

### 3.1. Study Selection

A total of 1531 studies were identified from the initial search, in which 16 RCTs [2,4,17,20,22,24,25,26,32,33,34,35,36,37,38,39] satisfied the eligibility criteria and were included in the current review (Figure 1) and ten [2,17,20,22,26,32,33,35,36,37] were included in the meta-analyses.

For the grey literature, we identified 81 studies from the initial search, including 49 RCT registers and 32 conference abstracts. After screening, we excluded these studies from the review due to wrong interventions or comparisons, wrong outcomes, not full text, or duplicate data.

### 3.2. Study Characteristics

Thirteen studies [2,17,20,22,24,25,26,32,33,35,36,37,39] evaluated the anti-inflammatory and anti-microbial effects, comparing the mouthwashes group to the placebos group with a total of 1961 participants; two studies [4,38] compared mouthwashes to no intervention with a total of 236 participants; one study [34] compared mouthwashes to usual care with 547 participants (Table 1). Twelve [2,4,17,20,22,24,25,26,32,35,38,39] of the included studies were conducted after the year 2010. In addition, the average age of the populations was over 45 years old, with the exception of one trial [17], which was carried out in a paediatric ICU. Three [25,33,39] of the included studies reported the rate of smokers.

The details of the interventions are provided in Table 1. The most commonly used mouthwash in the included studies was chlorhexidine, while some studies also reported mouthwashes with other chemical or herbal ingredients, e.g., sodium bicarbonate, hydrogen peroxide, and aloe vera extraction. All of the included trials used mouthwashes twice a day or more. Nine studies [2,17,22,25,32,34,35,36,39] used mouthwashes by swab, syringe, or foam, while other studies did not detail the device. Eight studies demonstrated that mouthwashes were effective on inflammation and bacteria compared to the placebo, e.g., by decreasing the incidence of VAP [22,33,35,37], the mortality [33], and the bacterial colonization rate [20,24,33,39], and improving the oral health status [25]. One study [4] demonstrated that mouthwashes were effective in decreasing the incidence of VAP and one [38] in decreasing the Clinical Pulmonary Infection Score when compared to the blank control group. One study [34] demonstrated that mouthwashes were not effective on inflammation and bacteria compared to usual care.

Fifteen studies [2,4,17,20,22,24,25,32,33,34,35,36,37,38,39] followed two or more days and the others did not detail the relevant information. Six studies [2,17,32,34,35,39] reported the ratio of loss was over 5%, which may be due to the wean of mechanical ventilation, and five studies [4,22,24,26,33] did not report the relevant information. One article [37] underlined the side effects of using mouthwash and reported oral mucosa irritation, and three studies [32,35,36] stated that there were no side effects. The remaining studies did not provide any information regarding the side effects of the mouthwashes.

### 3.3. Risk of Bias within Studies

The risk of bias assessed for the RCTs is summarized in Figure 2 and Figure 3. Six studies [24,32,34,36,37,38] included a high risk of bias and seven studies [2,4,25,26,33,35,39] included some concerns of a risk of bias. Three trials [34,36,38] introduced a high risk of bias in the selection of the reported result for multiple eligible outcome measurements within the outcome domain, two [24,38] in the measurement of the outcome for missing binding information, one [37] in the randomization process for the failure of allocating a sequence random, and one [32] in deviations from the intended interventions for the failure of blinding the participants and personnel.

### 3.4. Results of Individual Studies and Synthesis of Results

This review primarily focuses on studying the effects of using mouthwashes compared to placebo, no intervention, or usual care for primary outcomes (including the incidence of VAP, the mortality, and the duration of mechanical ventilation) and secondary outcomes (including the colonization of bacteria). Four studies [22,33,35,37] reported a significant difference in decreasing the incidence of VAP, one [33] in decreasing mortality, four [20,24,33,39] in decreasing the bacterial colonization rate, and one [25] in improving the oral health status when comparing mouthwashes to placebos. One trial [4] reported a significant difference in decreasing the incidence of VAP and one [38] in decreasing the Clinical Pulmonary Infection Score when comparing mouthwashes group to blank control group. In contrast, the other studies observed no significant difference between the experimental and control groups.

In consideration of reliability, we conducted meta-analyses only when the outcomes (including the incidence of VAP, the mortality, the duration of mechanical ventilation, and the colonization of pathogenic bacteria between mouthwashes and placebos groups) were reported by sufficient studies (*n* ≥ three).

#### 3.4.1. Meta-Analyses Comparing the Effect of Mouthwashes on VAP Incidence to Placebos

When evaluating the effect of mouthwashes compared to placebos on the incidence of VAP by meta-analyses commands, eight studies [2,17,22,32,33,35,36,37] were included (Figure 4a). Three studies [32,36,37] were evaluated as high risk. The incidence of VAP in the end of the follow-up was significantly lower in the mouthwashes group compared to the placebos group (OR 0.53, 95% CI 0.33 to 0.86, *p* = 0.01). In the meta-analyses, moderate heterogeneity was observed (Chi^2^ = 11.8 >7, I^2^ = 41%).

#### 3.4.2. Meta-Analyses Comparing the Effect of Mouthwashes on Mortality to Placebos

Four studies [17,32,35,36] were included to evaluate the clinical effect of mouthwashes compared to placebos on mortality (Figure 4b). Two studies [32,36] were evaluated as high risk. Although the mortality in the mouthwashes group was higher than the placebos group, no significant difference was observed (OR 1.49, 95% CI 0.92 to 2.40, *p* = 0.10). In the meta-analyses, low heterogeneity was observed (Chi^2^ = 2.57 < 3, I^2^ = 0%).

#### 3.4.3. Meta-Analyses Comparing the Effect of Mouthwashes on the Duration of Mechanical Ventilation to Placebos

Three studies [17,35,36] were included to evaluate the effect of mouthwashes compared to placebos on decreasing the duration of mechanical ventilation (Figure 4c). One study [36] was evaluated as high risk. No significant difference was observed in the duration of mechanical ventilation (WMD −0.10, 95% CI −2.01 to 1.81, *p* = 0.92) between the two groups. In the meta-analyses, moderate heterogeneity was observed (Chi^2^ = 5.49 > 2, I^2^ = 64%).

#### 3.4.4. Meta-Analyses Comparing the Effect of Mouthwashes on the Colonization of Bacteria to Placebos

When evaluating the effect of mouthwashes on the colonization of bacteria by meta-analyses commands, three studies [17,20,26] were included for *Staphylococcus aureus* (Figure 4d) analyses, four [2,17,26,32] for *Escherichia coli* (Figure 4e), and three [17,26,32] for *Pseudomonas aeruginosa* (Figure 4f). One study [32] was evaluated as high risk for the outcome of *Escherichia coli* and *Pseudomonas aeruginosa*. No significant difference was observed between the mouthwash and placebo groups in the number of subjects infected by *Staphylococcus aureus* (OR 0.88, 95% CI 0.34 to 2.30, *p* = 0.80), *Escherichia coli* (OR 1.19, 95% CI 0.50 to 2.82, *p* = 0.70), and *Pseudomonas aeruginosa* (OR 1.16, 95% CI 0.27 to 4.91, *p* = 0.84). In these meta-analyses, low heterogeneity was observed (*Staphylococcus aureus*: Chi^2^ = 1.06 < 2, I^2^ = 0%; *Escherichia coli*: Chi^2^ = 0.60 < 3, I^2^ = 0%; *Pseudomonas aeruginosa*: Chi^2^ = 1.94 < 2, I^2^ = 0%).

### 3.5. Results of Sensitivity Analysis

The results of the sensitivity analysis showed that the exclusion of any literature did not significantly affect the results of the meta-analyses (Figure 5).

### 3.6. Risk of Bias across Studies

The Egger’s test was used to assess the publication bias. No significant publication bias was detected in all of the meta-analyses (including the incidence of VAP (95% CI −3.89 to 5.53, *p* = 0.684), the mortality (95% CI −23.66 to 27.77, *p* = 0.764), the duration of mechanical ventilation (95% CI −177.85 to 162.76, *p* = 0.673), and the colonization of *Staphylococcus aureus* (95% CI −17.57 to 20.57, *p* = 0.500), *Escherichia coli* (95% CI −0.96 to 2.59, *p* = 0.186), and *Pseudomonas aeruginosa* (95% CI −19.91 to 14.79, *p* = 0.312)).

## 4. Discussion

In the current study, we found that mouthwashes might not be effective on the mortality, the duration of mechanical ventilation or the colonization of pathogenic bacteria for critical care patients, but they can significantly reduce the incidence of VAP. The results showed that mouthwashes can be used as a daily oral care supplement for critical care patients to control the incidence of VAP.

The study found that mouthwashes had a positive effect on decreasing the incidence of VAP for critical care patients. The pathogenesis of VAP is multifactorial, and substantial evidence indicated that pathogen colonization due to the poor oral hygiene could be one significant risk factor of VAP [3,11,40]. It was suggested that the active pharmaceutical ingredients within mouthwashes could integrate with bacterial components such as lipopolysaccharide and proteases, ultimately reducing the potential virulence of the bacteria [35,41]. Moreover, the mouthwashes could prevent the inhalation of secretions with pathogenic bacteria into the pulmonary tract by sterilizing the oropharynx, thus reducing the incidence of VAP [22]. In addition, oral hygiene with bactericide could improve the quality of life for critical care patients [42,43], which might provide essential support in the prevention of VAP.

No significant effect on mortality was observed in the mouthwash group compared to the placebo group in the current meta-analysis. A recent study found that mouthwashes might cause a nitric oxide-deficient condition owing to the destruction of oral bacterial flora, ultimately decreasing the bioavailability of nitric oxide [44]. The decrease in nitric oxide bioavailability might lead to multiple negative effects and has already been detected to have a relationship to the occurrence or deteriorating of high-mortality pathologies, including sepsis, diabetes and atherosclerosis [44]. Furthermore, patients in ICUs are usually older and their immune system is vulnerable [4]. Moreover, the majority of them have underlying diseases, such as hypertension and diabetes, which would increase the risk of mortality. Further studies need to pay more attention to decreasing the risk of death when considering mouthwashes as oral care for critical care patients.

A longer duration of mechanical ventilation might result in the accumulation of infectious agents [4]. Therefore, this has been suggested to be a risk factor for the progress of VAP by univariate analysis [4,17,45]. However, the present study found no significant difference in the duration of mechanical ventilation between the mouthwash and placebo groups with the limited evidence. More high-quality trials are needed to confirm or update the finding.

The main pathogenic bacteria of VAP include *Staphylococcus aureus*, *Escherichia coli*, and *Pseudomonas aeruginosa* [10,46,47]. However, no significant difference was observed in the colonization of the above bacteria between the mouthwash and placebo groups. The results indicated that dental plaque might be a major reservoir of pathogens [48,49]. The protective effects of dental plaque on both limiting the diffusion of mouthwashes and affecting their effects on bacteria might explain our findings [50]. Mouthwashes could reduce the potential virulence of bacteria [35,41] without killing bacteria and destroying normal oral flora, ultimately decreasing the incidence of VAP. According to the previous study, the destruction of normal oral flora might increase the risk of death [44]. The type, concentration, dosage, and frequency of use of mouthwashes might also influence the anti-bacterial effect. An article [51] observed that 2% chlorhexidine was more effective than a 0.2% solution in reducing the colonization of bacteria; however, no consensus has been achieved thus far on the usage of mouthwashes for patients in ICUs. In addition, an article [20] observed that 0.2% chlorhexidine was more effective than chamomile extract solution in reducing the colonization of bacteria.

A study [37] underlined the irritation of oral mucosa in individual participants during mouthwash use. Thus, the use of mouthwashes for critical care patients should be balanced against the unpleasant side effects. In addition, regardless of the mouthwash used, a thorough oral cleansing must be obtained to remove dental plaque, which is thought to be a stockpile of micro-organisms with pathogenic potential [2,52]. In the future, a comprehensive oral care unit containing mouthwashes is needed to achieve better clinical effects, which could play a crucial part in reducing the formation of dental plaque and the incidence of infections [53].

To our knowledge, the current systematic review is the first study to evaluate both the anti-inflammatory and anti-bacterial effects of mouthwashes as a part of daily oral care for patients admitted to ICUs. However, there were also some limitations that should be taken into consideration. First, the studies included in this review varied in hospital, patient demographics and comorbidities, patient oral hygiene status, the protocol of oral care, types of clinical outcome measures, and the length of follow-up; therefore, heterogeneity might exist. However, all of the included studies were RCTs and the baseline characteristics of the intervention and comparison groups within each trial were comparable; hence, collecting and compounding the specific clinical outcomes at the endpoint could compare the effects of mouthwashes to placebos. Second, six studies [24,32,34,36,37,38] were assessed with a high risk of bias due to the failure of the random process, deviations from the intended interventions, the measurement of the outcome, or the selection of the reported result. As a result, the internal validity of the study might be impaired. However, the results of the sensitivity analysis indicated that the exclusion of any literature did not significantly affect the results of the meta-analyses. Thus, the finding of this review could be regarded as relatively robust with a good degree of certainty. In addition, due to the limited number of included studies, we could not assess the effect of mouthwashes compared to no intervention and usual care, and we could not assess the oral inflammatory conditions and plaque index after the intervention of mouthwashes for patients in ICUs. Hence, more high-quality RCTs with a rigorous study design and large sample size are needed to evaluate both the anti-inflammatory and anti-bacterial effects of mouthwashes for critical care patients in order to strongly support clinical decision-making.

## 5. Conclusions

Patients in ICUs are at an increased risk of infection due to mechanical ventilation and poor oral hygiene. Although mouthwashes might not be effective on mortality, the duration of mechanical ventilation, and the colonization of pathogenic bacteria for critical care patients, they can significantly reduce the incidence of VAP. Thus, it suggested that mouthwashes can be used as a supplement to daily oral care for critical care patients in order to control the incidence of VAP. In order to support this finding further, more high-quality RCTs are needed to evaluate both the anti-inflammatory and anti-bacterial effects of mouthwashes for critical care patients.

## Figures and Tables

**Figure 1 ijerph-20-00733-f001:**
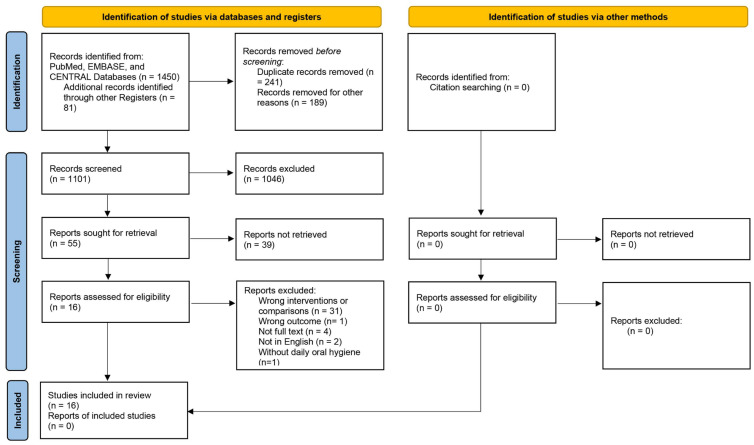
Preferred Reporting Items for Systematic Reviews and Meta-Analyses (PRISMA) flow diagram for study selection (CENTRAL, Cochrane Central Register of Controlled Trials).

**Figure 2 ijerph-20-00733-f002:**
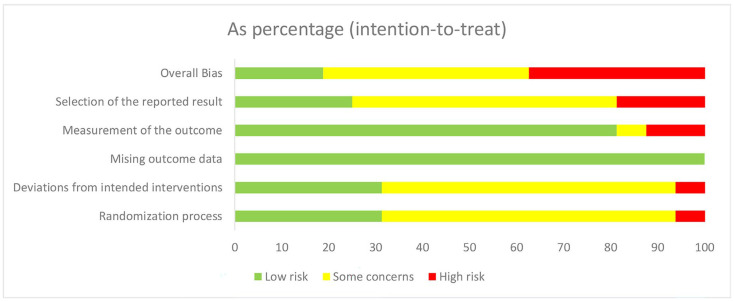
The risk of bias graph.

**Figure 3 ijerph-20-00733-f003:**
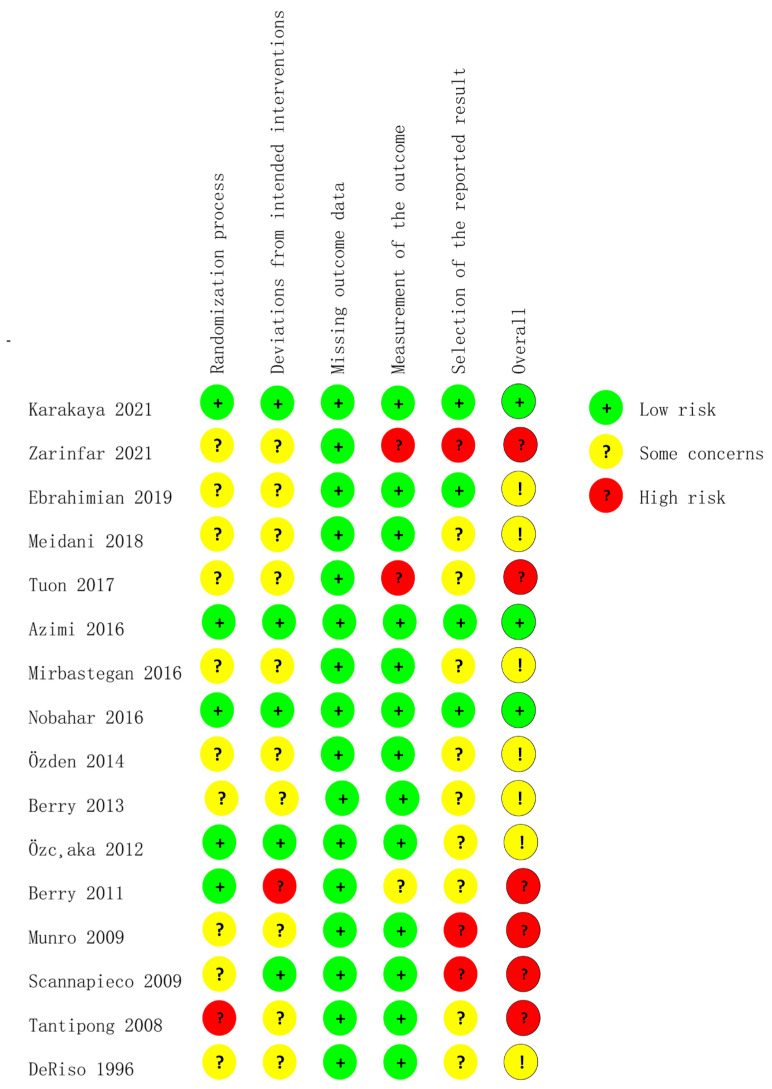
The risk of bias summary [2,4,17,20,22,24,25,26,32,33,34,35,36,37,38,39].

**Figure 4 ijerph-20-00733-f004:**
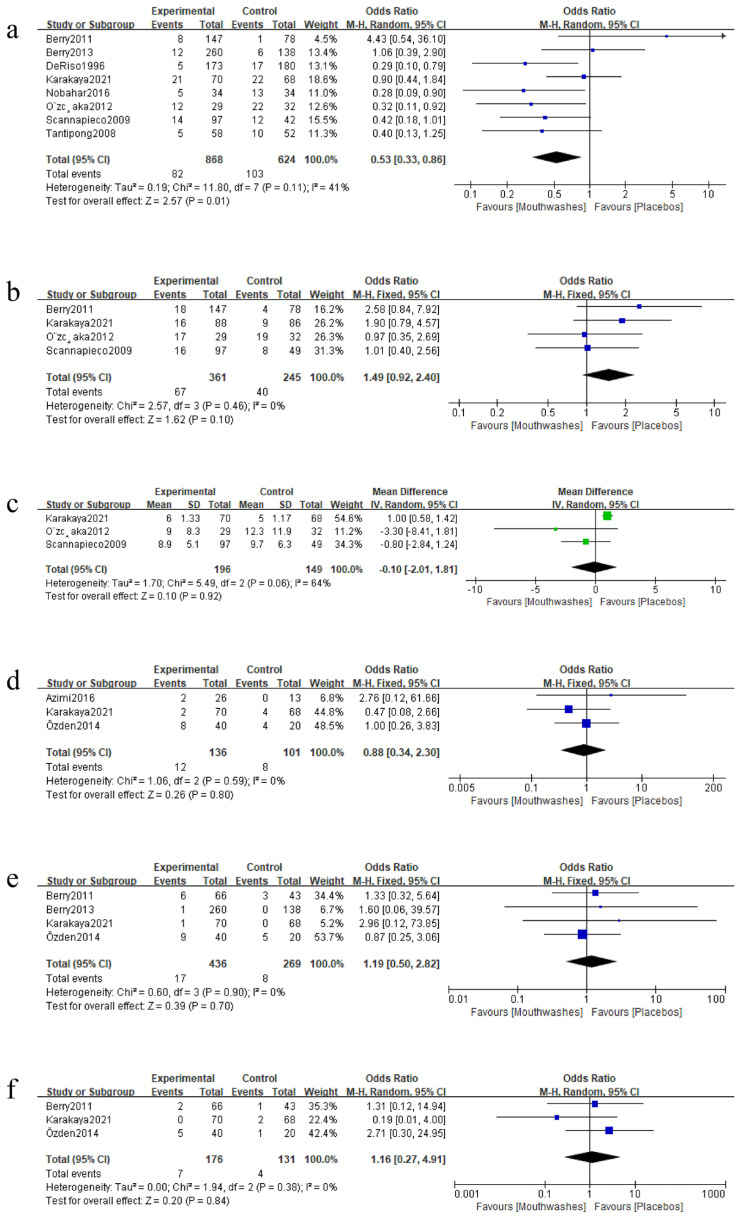
Forest plots comparing the primary and secondary outcomes between mouthwashes and placebos. Comparisons for primary outcomes (including (**a**) the incidence of ventilator associated pneumonia [2,17,22,32,33,35,36,37], (**b**) the mortality [17,32,35,36], and (**c**) the duration of mechanical ventilation [17,35,36]) and secondary outcomes (including the colonization of (**d**) *Staphylococcus aureus* [17,20,26], (**e**) *Escherichia coli* [2,17,26,32], and (**f**) *Pseudomonas aeruginosa* [17,26,32]). The blue square represents odds ratios and the green square stands for the weighted mean differences.

**Figure 5 ijerph-20-00733-f005:**
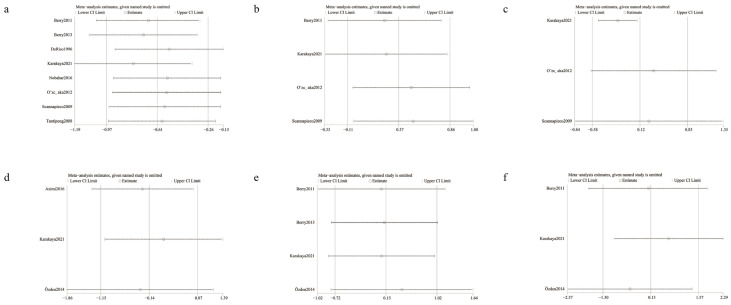
The results of sensitivity analysis (including (**a**) the incidence of ventilator associated pneumonia [2,17,22,32,33,35,36,37], (**b**) the mortality [17,32,35,36], (**c**) the duration of mechanical ventilation [17,35,36], and the colonization of (**d**) *Staphylococcus aureus* [17,20,26], (**e**) *Escherichia coli* [2,17,26,32], and (**f**) *Pseudomonas aeruginosa* [17,26,32]).

**Table 1 ijerph-20-00733-t001:** The characteristics of qualified studies.

Study	Sample Size	Average Age (Year)	Males (%)	Smoker (%)	Mouthwash(es) ^#^	Comparison(s) ^#^	Devices for Applying Mouthwashes	Relevant Clinical Measures	Effects	Follow-up	Loss (%)	Side Effects
Karakaya et al., 2021 * [17]	174	45 (month)	112 (64.4%)	NR	0.12% CHX,5 mL, six times a day	Placebo, 5 mL, six times a day	Swab	VAP; Mortality; Duration of MV; Duration of ICU stay; Duration of hospital stay; Colonization of bacteria	No significant difference was observed on the above outcome.	14 days	39 (22.4%)	NR
Zarinfar et al., 2021 [38]	86	54.6 ±21.8	66 (51.2%)	NR	?% CHX, ? ml, twice daily	Blank control	NR	VAP; CPIS	Significant difference was observed in CPIS. No significant difference was observed in the incidence of VAP.	NR	0	NR
Ebrahimian et al., 2019 [25]	90	52.35	57 (63.3%)	28 (31.1%)	① 0.02% Zufa, 15 mL, twice a day; ② 2% CHX, 15 mL, twice a day	0.9% NaCl, 15 mL, twice a day	Syringe	BOAS	Significant difference was observed on the BOAS comparing experiment group to control group.	3 days	2 (2.2%)	NR
Meidani et al., 2018 [4]	150	50.7 ± 20.2	107 (71.3%)	NR	① 0.2% CHX, ? ml, three times a day; ② 0.01% potassium permanganate, ? ml, three times a day	Blank control	NR	VAP; Mortality	Significant difference was observed in VAP among the three groups. No significant difference was observed in mortality among the three groups.	1 week	NR	NR
Tuon et al., 2017 [24]	16	47.9	9 (56%)	NR	2% CHX,15 mL, twice daily	0.9% NaCl, 15 mL, twice daily	NR	Colonization of bacteria	Significant difference was observed on the colonization of bacteria.	10 days	NR	NR
Azimi et al., 2016 [20]	39	45.97	21 (53.8%)	NR	① 0.2% CHX, 10 mL, three times per day; ② ?% matrica, 10 mL, three times per day	? % NaCl, 10 mL, three times per day	NR	Colonization of bacteria	Significant difference was observed on the bacterial colonization rate comparing CHX group to matrica and saline group.	2 days	0	NR
Mirbastegan et al., 2016 [39]	90	47.43	59 (65.6%)	32 (35.6%)	3% aloe vera, 15 mL, twice a day	0.9% NaCl, 15 mL, twice a day	Syringe	PI	Significant difference was observed on mean and standard deviation for dental PI.	4 days	11 (12.2%)	NR
Nobahar et al., 2016 [22]	68	64.7	34 (50%)	NR	3% hydrogen peroxide, 15 mL, twice daily	0.9% NaCl, 15 mL, twice daily	Swab	VAP	Significant reduction was observed on the incidence of VAP comparing hydrogen peroxide to normal saline groups.	5 days	NR	NR
Özden et al., 2014 [26]	60	61.6% were over 66 years old	34 (56.6%)	NR	① 5% sodium bicarbonate, ? ml, 3 times a day; ② 0.2% CHX, ? ml, 3 times a day	0.9% NaCl, ? ml, 3 times a day	NR	Oral assessment tool scores; Colonization of bacteria	No significant difference was observed on oral assessment tool scores among the three groups.	4 days	NR	NR
Berry et al., 2013 [2]	398	57.9	236 (59.3%)	NR	① ?% sodium bicarbonate, 20 mL, every two hours; ② ?% Listerine, 20 mL, twice a day and sterile water every 2 h	Sterile water, 20 mL, every two hours	Syringe	VAP; Duration of MV; Duration of ICU stay; Colonization of bacteria	No significant difference was observed in the duration of MV and ICU stay, and the colonization of bacteria.	4 days	45 (11.3%)	NR
Özc¸aka et al., 2012 [35]	66	58.1	NR	NR	0.2% CHX, 30 mL, four times a day	? % NaCl, 30 mL, four times a day	Swab	VAP; Mortality; Duration of MV; Duration of ICU stay; Colonization of bacteria	Significant reduction was observed on the rate of VAP comparing CHX to saline group. No significant difference was observed on the duration of MV and the mortality between two groups.	14 days	5 (7.6%)	0
Berry et al., 2011 [32]	225	59.2	121 (53.7%)	NR	① Sodium bicarbonate, ? ml, second hourly; ② 0.2% CHX, ? ml, twice daily and sterile water, ? ml second hourly	Sterile water rinsed, ? ml, second hourly	Syringe	VAP; Colonization of bacteria	No significant difference was observed on the colonization of bacteria among groups.	4 days	116 (51.6%)	0
Munro et al., 2009 [34]	547	47.9 ± 17.5	328 (60%)	NR	0.12% CHX,5 mL, twice daily	Usual care	Swab	VAP; CPIS	No significant difference was observed on CPIS and pneumonia between two groups.	7 days	362 (76.9%)	NR
Scannapieco et al., 2009 [36]	175	48.0	104 (71%)	NR	① 0.12% CHX, 30 mL, twice daily; ② 0.12% CHX, 30 mL, once daily, and placebo, 30 mL, once daily	Placebo, 30 mL, twice daily	Oral foam applicator	VAP; CPIS; Mortality; Duration of MV; Duration of ICU stay; Colonization of bacteria; Plaque score	No significant reduction was observed on the colonization of *Staphylococcus aureus*, the incidence of VAP comparing CHX to placebo groups.	21 days	0 (0%)	0
Tantipong et al., 2008 [37]	207	58.3	101 (48%)	NR	2% CHX, 15 mL, 4 times per day	0.9% NaCl,15 mL, 4 times per day	NR	VAP; Colonization of bacteria	Significant reduction was observed on the incidence of VAP comparing CHX to saline group.	2 days	0 (0%)	Irritation of the oral mucosa
DeRiso et al., 1996 [33]	353	63.8	242 (68.6%)	208 (58.9%)	0.12% CHX, ? ml, twice-daily	Placebo, ? ml, twice-daily	NR	VAP; Mortality; Duration of MV; Colonization of bacteria	Significant reduction was observed in the incidence of VAP, mortality, and the colonization of bacteria comparing CHX to placebo groups. No significant difference was found on the duration of MV between groups.	NR	NR	NR

Note: NR, not reported; CHX, chlorhexidine; NaCl, normal saline; 1. Inflammatory related indices: VAP: ventilator associated pneumonia; CPIS: Clinical Pulmonary Infection Score by Luna 2003; Mortality; Duration of mechanical ventilatory (MV); Duration of intensive care unit (ICU) stay; Duration of hospital stay; BOAS: Beck Oral Assessment Scale by Beck 1979; Oral assessment tool scores; 2. Bacterial related indices: Colonization of bacteria; Plaque score; PI: Plaque Index; 3. Different mouthwashes applied in the same study were marked as ① and ②. * The study marked with grey was carried out in paediatric intensive care unit (PICU); ^#^ The unreported details of mouthwashes in selected studies were described with question marks (e.g., ? mL).

## Data Availability

The data are contained within the article.

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
