# Peer review of "Anti-Inflammatory and Anti-Bacterial Effects of Mouthwashes in Intensive Care Units: A Systematic Review and Meta-Analysis"

_ijerph, 2022, doi:10.3390/ijerph20010733_

Round 1

Reviewer 1 Report

The authors intended to evaluate the effects of mouthwashes in ICU patients in this systemic review. 

However, the selected papers did not satisfy the requirements.  In other words, some studies described the effects of cleansing or swabbing  but not so-called mouthwashes. Mouthwash means rinsing or gargling.

Author Response

Dear reviewer,

We are truly grateful to your critical comments and thoughtful suggestions on our manuscript titled ‘Anti-inflammatory and antibacterial effects of mouthwashes for the critical care patients: a systematic review and meta-analysis’ (Manuscript ID: ijerph-2080085).

Based on these comments and suggestions, we have made careful modifications on the original manuscript. All changes made to the text are marked up using the “Track Changes” function.

Below you will find our point-by-point responses to your comments or questions. Responses are indicated by ‘Author response’ and revisions outlined under ‘Author action’ below the reviewer’s comment.

1.1 Reviewer comment

However, the selected papers did not satisfy the requirements. In other words, some studies described the effects of cleansing or swabbing but not so-called mouthwashes. Mouthwash means rinsing or gargling.

1.1 Author response

Thank you very much for your comment.

The participants of the selected studies in our review were identified as patients admitted to ICUs with endotracheal tubes or mechanical ventilation, for whom it’s difficult to maintain daily oral hygiene including tooth brushing or even by rinsing or gargling [1]. Instead, nursers would also use mouthwashes by swab, syringe, or foam, limited by endotracheal tubes or mechanical ventilation, to control the inflammation and infection for ICU patients [2-7]. Thus, we also included studies on the effects of mouthwashes by swab, syringe, or foam among ICU patients.

We had acknowledged that the heterogeneity across the methodology might exist in discussion section, Line 355-360. However, all the included studies were RCTs and the baseline characteristics of the intervention and comparison groups within each trial were comparable; hence, collecting and compounding the specific clinical outcomes at the endpoint could compare the effects of mouthwashes to placebos. And the results of sensitivity analysis indicated that the exclusion of any literature did not significantly affect the results of meta-analyses. Thus, the finding of this review could be regarded as relatively robust with a good degree of certainty, and it could provide comprehensive evidence for clinical decision making and practice on the use of mouthwashes among critical care patients.

1.1 Author action

No action taken.

Reviewer 2 Report

Dear Authors,

Mouthwashes or dentifrices are supplementary products in oral hygiene! This topic is original and relevant!

Some suggestions should be performed: 

The authors should perform detailed consideration regarding the type of dentifrices (such as with or without chlorhexidine, only reference 46 is highlighted?) and the device or their implementation in dental practice! Line 304, “the risk of death,” should be engaged in the sentence. Lines 313-314 state the protective effect of dental plaque … it is a controversial idea… the authors should explain and correlate it into the scope of the manuscript. 

Lines 71 to 75 can be considered eligible criteria 

The devices used to apply the mouthwashes should be detailed 

The research strategy timeline (beginning and end) should be clear (Lines 97-98)

Why did the authors not perform a scoping review of the grey literature that was studied? And the websites should be presented (lines 104-110). 

In figure 1, the identification of studies via other methods should be clarified! 

The table 1, the legend should be detailed (namely ① … symbol) 

Figure 5 is too difficult to follow. 

Study limitation and bias are highlighted (lines 328-344), and the authors should consider the type of rehabilitation that engage a different kind of oral hygiene, such as removable or fixed rehabilitation, and comorbidities, such as diabetes, and the strategy of oral hygiene. Mechanical strategy is the core of clinical motivation!

Author Response

Response to Reviewer 2 Comments

Dear reviewer,

We are truly grateful to your critical comments and thoughtful suggestions on our manuscript titled ‘Anti-inflammatory and antibacterial effects of mouthwashes for the critical care patients: a systematic review and meta-analysis’ (Manuscript ID: ijerph-2080085).

Based on these comments and suggestions, we have made careful modifications on the original manuscript. All changes made to the text are marked up using the “Track Changes” function.

Below you will find our point-by-point responses to your comments or questions. Responses are indicated by ‘Author response’ and revisions outlined under ‘Author action’ below the reviewer’s comment.

2.1 Reviewer comment

The authors should perform detailed consideration regarding the type of dentifrices (such as with or without chlorhexidine, only reference 46 is highlighted?) and the device or their implementation in dental practice! Line 304, “the risk of death,” should be engaged in the sentence.

2.1 Author response

Thank you very much for your suggestion.

According to your suggestion, we added reference comparing chlorhexidine mouthwash to chamomile extract solution in Discussion Section. And we also detailed the devices for applying mouthwashes in Table 1. However, it has been reported that no significant difference was observed on the colonization of bacteria, when comparing different devices for applying mouthwashes [1]. Moreover, we had excluded the potential heterogeneity from different devices for applying mouthwashes by sensitivity analysis. We added the following details in the manuscript.

2.1 Author action

Discussion section, Line 337-348: It indicated that dental plaque might be a major reservoir of pathogens [48, 49]. The protective effects of dental plaque on both limiting the diffusion of mouthwashes and affecting their effects on bacteria might explain our findings [50]. While mouthwashes could reduce the potential virulence of bacteria [35, 41] without killing bacteria and destroying normal oral flora, ultimately decreasing the incidence of VAP. According to the previous study, the destroy of normal oral flora might increase the risk of death [44]. The type, concentration, dosage and frequency of mouthwashes might also influence the antibacterial effect. An article [51] observed that 2% chlorhexidine was more effective than 0.2% solution on reducing the colonization of bacteria, though no consensus has been achieved by far on the usage of mouthwashes for patients in ICUs. In addition, an article [20] observed that 0.2% chlorhexidine was more effective than chamomile extract solution on reducing the colonization of bacteria.

2.2 Reviewer comment

Lines 313-314 state the protective effect of dental plaque … it is a controversial idea… the authors should explain and correlate it into the scope of the manuscript.

2.2 Author response

Thank you very much for your suggestion.

It has been reported that dental plaque could both limit the diffusion of mouthwashes and affect the effects of mouthwashes on bacteria [2]. Thus, we stated the protective effect of dental plaque, which protected bacteria from mouthwash. It might be not clear in our words.

We added the following details in the manuscript to make it clearer.

2.2 Author action

Discussion section, Line 337-343: It indicated that dental plaque might be a major reservoir of pathogens [48, 49]. The protective effects of dental plaque on both limiting the diffusion of mouthwashes and affecting their effects on bacteria might explain our findings [50]. While mouthwashes could reduce the potential virulence of bacteria [35, 41] without killing bacteria and destroying normal oral flora, ultimately decreasing the incidence of VAP.

2.3 Reviewer comment

Lines 71 to 75 can be considered eligible criteria.

2.3 Author response

Thank you very much for your suggestion. Lines 71 to 75 were Population-Interventions-Comparisons-Outcomes-Study Design (PICOS) elements of the study, and we detailed eligible criteria from Line 81 to 99.

2.3 Author action

No action taken.

2.4 Reviewer comment

The devices used to apply the mouthwashes should be detailed.

2.4 Author response

Thank you very much for your suggestion. We added the following details in the manuscript.

2.4 Author action

Table 1: relevant information was detailed on the column of Devices for applying mouthwashes.

Results section, line 197-199: Nine studies [2, 17, 22, 25, 32, 34-36, 39] used mouthwashes by swab, syringe, or foam, while other studies did not detail the devices.

2.5 Reviewer comment

The research strategy timeline (beginning and end) should be clear (Lines 97-98)

2.5 Author response

Thank you very much for your suggestion. We had detailed the research strategy timeline at Line 101-103: PubMed, EMBASE, and Cochrane Central Register of Controlled Trials (CENTRAL) databases were systematically searched for relevant articles written in English, from inception to 29 March 2022.

2.5 Author action

No action taken.

2.6 Reviewer comment

Why did the authors not perform a scoping review of the grey literature that was studied? And the websites should be presented (lines 104-110).

2.6 Author response

Thank you very much for your suggestion. The scoping of grey literature had also performed based on the search strategy, combining population (intensive care unit patients) and intervention (mouthwashes). We added the details in the Result Section.

Also, we detailed the websites of grey literature in methods section.

2.6 Author action

Methods section, line 110-119: For unpublished works, “grey” literature databases were also searched following the above search strategy, combining population (intensive care unit patients) and intervention (mouthwashes). For unpublished clinical trials or registries, the ClinicalTrials.gov (https://beta.clinicaltrials.gov/) and the International Clinical Trials Registry Platform (https://www.who.int/clinical-trials-registry-platform) were searched. For relevant dissertations and theses, the ProQuest Dissertation Abstracts and Thesis database (https://about.proquest.com/) was searched. The Web of Science (http://www.webofscience.com) was searched by the Conference Proceedings Citation Index-Science for conference proceedings. Additionally, some other online resources such as the GreyNet databases (https://easy.dans.knaw.nl/ui/home) were also searched as a supplement.

Result section, line 178-180: For grey literature, we identified 81 studies from the initial search, including 49 RCT registers and 32 conference abstracts. After screening, we excluded these studies from the review for wrong interventions or comparisons, wrong outcomes, not full text, or duplicate data.

2.7 Reviewer comment

In figure 1, the identification of studies via other methods should be clarified!

2.7 Author response

Thank you very much for your suggestion. In the part of the identification of studies via other methods, we identified studies by citation search. We added the following details in Figure 1.

2.7 Author action

Figure 1, identification of studies via other methods:

Records identified from:

Citation searching (n = 0).

2.8 Reviewer comment

The table 1, the legend should be detailed (namely ① … symbol)

2.8 Author response

Thank you very much for your suggestion. We added the following details in the legend of Table 1.

2.8 Author action

The legend of Table 1, Line 214-221:

Note: NR, not reported; CHX, chlorhexidine; NaCl, normal saline;

  1. Inflammatory related indices: VAP: ventilator associated pneumonia; CPIS: Clinical Pulmonary Infection Score by Luna 2003; Mortality; Duration of mechanical ventilatory (MV); Duration of intensive care unit (ICU) stay; Duration of hospital stay; BOAS: Beck Oral Assessment Scale by Beck 1979; Oral assessment tool scores;
  2. Bacterial related indices: Colonization of bacteria; Plaque score; PI: Plaque Index;
  3. Different mouthwashes applied in the same study were marked as ① and ②.

*The study marked with grey was carried out in paediatric intensive care unit (PICU);

#The unreported details of mouthwashes in selected studies were described with question marks (e.g., ? mL).

2.9 Reviewer comment

Figure 5 is too difficult to follow.

2.9 Author response

Thank you very much for your comment. The size of figure might be compressed. We uploaded the figure separately.

2.9 Author action

Figure file, Figure 5 has been submitted.

2.10 Reviewer comment

Study limitation and bias are highlighted (lines 328-344), and the authors should consider the type of rehabilitation that engage a different kind of oral hygiene, such as removable or fixed rehabilitation, and comorbidities, such as diabetes, and the strategy of oral hygiene. Mechanical strategy is the core of clinical motivation!

2.10 Author response

Thank you very much for your suggestion. We added the following details in the discussion section.

2.10 Author action

Discussion section, line 359-364: First, the included studies of this review varied in hospital, patient demographics and comorbidities, patient oral hygiene status, the protocol of oral care, types of the clinical outcome measures, and the length of follow-up, where the heterogeneity might exist. However, all the included studies were RCTs and the baseline characteristics of the intervention and comparison groups within each trial were comparable; hence, collecting and compounding the specific clinical outcomes at the endpoint could compare the effects of mouthwashes to placebos.

Reviewer 3 Report

Xiang Y et al. conducted a meta-analysis about anti-inflammatory and antibacterial effects of mouthwashes in intensive care units. The protocol was well designed and the readers for IJERPH will have an interest in this topic.

However, the content of the discussion was unsatisfactory despite detailed analysis.

Major comments:

1)    L326- To our knowledge, the current systematic review is the first study to evaluate the anti-inflammatory and antibacterial effects of mouthwashes as a part of daily oral care for patients admitted to ICUs.

This is not true. How about the following reviews?

a)    Zhan T et al. Cochrane Database Syst Rev. 12(12): CD008367.

b)    Rabello F et al. Int J Dent Hyg 16(4): 441-449, 2018.

2)    L56- no systemic review comprehensively evaluated the anti-inflammatory and antibacterial effects of mouthwashes for critical care patients, especially on the anti-inflammatory and antibacterial effects of mouthwashes for critical care patients.

 This is true. However, the reviewer could not understand the significance of analyzing each pathogenic bacterium. Because the mouthwash has a broad antibacterial spectrum, not bacteria-specific action.

And the word “systemic” is probably “systematic”

3)    Significant difference was found in the incidence of VAP. However, there was no significant difference regarding as the colonization of each pathogenic bacteria. So what factors suppressed the incidence of VAP?

Minor comments:

1)    Figure 5 was vague due to the low resolution.

2)    The authors are recommended to check the grammar and the spells throughout the manuscript. For example, Ozc,aka et al., 2012 in Table 1 is wrong. CHX is chlorhexidine gluconate.

Author Response

Response to Reviewer 3 Comments

Dear reviewer,

We are truly grateful to your critical comments and thoughtful suggestions on our manuscript titled ‘Anti-inflammatory and antibacterial effects of mouthwashes for the critical care patients: a systematic review and meta-analysis’ (Manuscript ID: ijerph-2080085).

Based on these comments and suggestions, we have made careful modifications on the original manuscript. All changes made to the text are marked up using the “Track Changes” function.

Below you will find our point-by-point responses to your comments or questions. Responses are indicated by ‘Author response’ and revisions outlined under ‘Author action’ below the reviewer’s comment.

3.1 Reviewer comment

L326- To our knowledge, the current systematic review is the first study to evaluate the anti-inflammatory and antibacterial effects of mouthwashes as a part of daily oral care for patients admitted to ICUs.

This is not true. How about the following reviews?

Zhan T et al. Cochrane Database Syst Rev. 12(12): CD008367.

Rabello F et al. Int J Dent Hyg 16(4): 441-449, 2018.

3.1 Author response

Thank you very much for your suggestion. There were some meta-analyses have explored the efficacy of chlorhexidine mouthwash on preventing the VAP [1, 2], but no systematic review comprehensively evaluated the antibacterial effects of mouthwashes for critical care patients. Zhan T et al. [1] evaluated the effect of chlorhexidine, but not mouthwashes. Rabello F et al. [2] only evaluated anti-inflammatory effects of chlorhexidine mouthwash. Thus, we considered that the current systematic review is the first study to evaluate the antibacterial effects of mouthwashes as a part of daily oral care for patients admitted to ICUs.

It might be not clear in our words. We revised the sentence in the manuscript.

3.1 Author action

Discussion section, line 356-358: To our knowledge, the current systematic review is the first study to evaluate both the anti-inflammatory and antibacterial effects of mouthwashes as a part of daily oral care for patients admitted to ICUs.

3.2 Reviewer comment

L56- no systemic review comprehensively evaluated the anti-inflammatory and antibacterial effects of mouthwashes for critical care patients, especially on the anti-inflammatory and antibacterial effects of mouthwashes for critical care patients.

This is true. However, the reviewer could not understand the significance of analyzing each pathogenic bacterium. Because the mouthwash has a broad antibacterial spectrum, not bacteria-specific action.

And the word “systemic” is probably “systematic”

3.2 Author response

Thank you very much for your suggestion.

Although mouthwash has a broad antibacterial spectrum, some studies have indicated that mouthwash was effective on reducing the colonization of specific bacteria, e.g., Staphylococcus aureus [3, 4], which was the main pathogenic bacteria of VAP [5]. Thus, we evaluated the colonization of each pathogenic bacterium to study the antibacterial effect of mouthwashes for ICU patients, so as to provide relevant evidence for clinical decision making and practice on patient critical care. We have added the following information to the Introduction section.

We had also reported the clinical outcome of plaque index or score in the Table 1. However, we could not evaluate the relevant outcomes by meta-analysis, limited by the number of included studies.

We have checked and revised the grammar and the spells throughout the manuscript.

3.2 Author action

Introduction section, line 54-55: And studies have indicated that mouthwashes were effective on reducing the incidence of VAP [4, 22] and the colonization of Staphylococcus aureus [23, 24].

Discussion section, line 370-373: Also, limited by the number of included studies, we could not assess the effect of mouthwashes compared to no intervention and usual care, and we could not assess the oral inflammatory conditions and plaque index, after the intervention of mouthwashes for patients in ICUs.

3.3 Reviewer comment

Significant difference was found in the incidence of VAP. However, there was no significant difference regarding as the colonization of each pathogenic bacteria. So what factors suppressed the incidence of VAP?

3.3 Author response

Thank you very much for your comment.

The pathogenesis of VAP is multifactorial, and substantial evidence indicated that the pathogen colonization due to the poor oral hygiene could be one significant risk factor of VAP [6-8]. Although no significant difference was observed on the colonization of regarding bacteria between mouthwash and placebo groups, it was suggested that the active pharmaceutical ingredients within mouthwashes could integrate with bacterial components such as lipopolysaccharide and proteases, ultimately reducing the potential virulence of the bacteria [9, 10]. It was also the advantage of mouthwashes which only reduce the potential virulence without killing bacteria and destroying normal oral flora. However, we could only evaluate the colonization of Staphylococcus aureus; Escherichia coli; and Pseudomonas aeruginosa, limited by the number of included studies. And we had acknowledged that mouthwashes might suppress the incidence of VAP by other ways from Line 308 to 312.

We added the following details in the manuscript.

3.3 Author action

Discussion section, line 340-342: While mouthwashes could reduce the potential virulence of bacteria [35, 41] without killing bacteria and destroying normal oral flora, ultimately decreasing the incidence of VAP.

3.4 Reviewer comment

Figure 5 was vague due to the low resolution.

3.4 Author response

Thank you very much for your comment. The size of figure might be compressed. We uploaded the figure separately.

3.4 Author action

Figure file, Figure 5 has been submitted.

3.5 Reviewer comment

The authors are recommended to check the grammar and the spells throughout the manuscript. For example, Ozc,aka et al., 2012 in Table 1 is wrong. CHX is chlorhexidine gluconate.

3.5 Author response

Thank you very much for your suggestion. We have checked and revised the grammar and the spells throughout the manuscript. We checked the article [9] and modified Ozc,aka et al. 2012 as Özc¸aka et al., 2012 in Table 1 and Figure 3. And generally, CHX stands for chlorhexidine in most studies related to mouthwashes [4, 11].

3.5 Author action

We have checked and revised the grammar and the spells throughout the manuscript. Table 1 and Figure 3: we checked the article and modified Ozc,aka et al. 2012 as Özc¸aka et al., 2012.

Round 2

Reviewer 1 Report

The manuscript has been revised appropriately.

Reviewer 3 Report

The authors responded to the reviewer's questions point by point in manner and revised the manuscript.